# CONTRASTIVELY ENFORCING DISTINCTIVENESS FOR MULTI-LABEL CLASSIFICATION

## ABSTRACT

Recently, as an effective way of learning latent representations, contrastive learning has been increasingly popular and successful in various domains. The success of contrastive learning in single-label classifications motivates us to leverage this learning framework to enhance distinctiveness for better performance in multi-label image classification. In this paper, we show that a direct application of contrastive learning can hardly improve in multi-label cases. Accordingly, we propose a novel framework for multi-label classification with contrastive learning in a fully supervised setting, which learns multiple representations of an image under the context of different labels. This facilities a simple yet intuitive adaption of contrastive learning into our model to boost its performance in multi-label image classification. Extensive experiments on two benchmark datasets show that the proposed framework achieves state-of-the-art performance in the comparison with the advanced methods in multi-label classification.

## 1 INTRODUCTION

Multi-label image classification is a fundamental and practical computer vision task, where the goal is to predict a set of labels (e.g., objects or attributes) associated with an input image. It is an essential component in many applications such as recommendation systems (Jain et al., 2016; Yang et al., 2015), medical image diagnosis (Ge et al., 2018b), and human attribute recognition (Li et al., 2016b). Compared to single-label cases, multi-label classification is usually more complex and challenging.

Recently, contrastive learning (CL) (Chen et al., 2020; He et al., 2020; Li et al., 2020; Caron et al., 2020; Bachman et al., 2019) has been shown as an effective pretext approach to learn latent representations an unsupervised way, which can be further used for supervised tasks. In general, CL aims to pull together an anchor and a similar (or positive) sample in embedding space and push apart the anchor from many dissimilar (or negative) samples. Therefore, the choice of the positive and negative samples of an anchor is a key to achieving good performance with CL. In self-supervised CL (Chen et al., 2020), the positive sample is defined as those augmented from the same image with the anchor, while the negative samples are all the other images in the minibatch. More recently, supervised CL (Khosla et al., 2020; Sun et al., 2021; Yuan et al., 2021; Huynh, 2021) has been proposed, where all the images with the same label as the anchor are considered as the positive samples and vice versa for the negative ones. Supervised CL has shown improvements in single-label image classifications than the self-supervised counterpart. With the above successful examples, CL has drawn significant research attention and has been applied in other tasks including image segmentation (Wang et al., 2021), adversarial training (Kim et al., 2020), and text to image learning (Radford et al., 2021).

Given the appealing properties and promising results of CL in single-label classification, it is natural to adapt it into multi-label cases to boost performance. However, this adaptation is non-trivial. In single-label cases, an image usually contains one salient object, thus, the label of the object can also be viewed as the unique label of the image. Therefore, it is reasonable to use one image-level representation of an image and to push the representation of the anchor close to its positive samples (e.g, augmentations of the anchor or images with the same label as the anchor), as done in self-supervised and supervised CL. However, with a single image-level representation for an image, it is hard to define the positive or negative samples for an anchor image by its multiple labels in the

multi-label classification. For example, it would not be reasonable to assume that the image-level representations of images containing apples must always be close to each other, as apple is just one of many objects in those images and an apple may only take a small area of an image. As a result, this setting hinders the application of existing CL methods to multi-label classification.

To bridge this gap, we in this paper propose a novel end-to-end framework for multi-label image classification that leverages the contrastive learning principle, termed **MulCon** or *Multi-label Classification with Contrastive Loss*. Instead of using image-level representations as in previous CL methods, we introduce a new module that learns multiple label-level representations of an image, which are generated with the attention from the globally class-specific embeddings to the image features learned by a convolutional neural network (CNN). Each label-level embedding of an image corresponds to the image's representation in the context of a specific label. With these definitions, the supervised CL loss can be applied. Specifically, if we look at one specific label and view a label-level embedding of an image as the anchor in our proposed CL framework, it is straightforward to define the positive samples of an anchor, which are the label-level embeddings of the other images in the minibatch with the same label and vice versa for the negative samples. For example, instead of sharing an image-level representation with other objects, the apple object of an image has its own embedding and the embeddings of the apples of all the images in a minibatch are pushed close to each other. Therefore, our framework is intuitive in the multi-label setting. In this way, the CL loss can enforce the coherence and consistency of the label-level representations of images, which further provides more discriminative power of the prediction procedure based on these representations.

The main contributions can be summarized as follows: **1)** Contrastive learning has been shown successful in many single-label classification problems, however, it is non-trivial to apply it in multi-label classification. We propose an intuitive and conceptually simple framework that encompasses contrastive learning. **2)** Along with the proposed framework, we also introduce a practical training scheme of contrastive learning for multi-label classification. **3)** We conduct extensive experiments on large-scale benchmark datasets, showing that the proposed framework achieves the state-of-the-art performance in multi-label image classification.

## 2 PROPOSED METHOD

In this section, we first discuss about the background knowledge and relevant notations and then elaborate on the details of the proposed framework, MulCon.

### 2.1 BACKGROUND AND NOTATIONS

**Multi-label Classification** Following the standard setting of multi-label image classification, we denote a minibatch of input images by $X \in \mathbb{R}^{N \times W \times H \times 3}$, where $N$ is the batch size, $H$ and $W$ are the height and width of the images. Each image $x_i \in X$ is associated with multiple labels selected from a set of $L$ labels in total, which are denoted by a multi-hot binary vector $y_i \in \{0, 1\}^L$. For an active label $j$ of $x_i$, $y_{ij} = 1$ and vice versa. Our task is to build an end-to-end model that takes $x_i$ to predict its labels $y_i$.

**Attention** The Attention mechanism (Luong et al., 2015; Xu et al., 2015) has been widely used in various areas of computer vision and natural language processing, which enhances the important parts of the data of interest and fades out the rest. Assume that $n_q$ query vectors of size $d_q$ denoted as $Q \in \mathbb{R}^{n_q \times d_q}$, and $n_v$ key-value pairs denoted as $K \in \mathbb{R}^{n_v \times d_q}$, $V \in \mathbb{R}^{n_v \times d_v}$. The attention function maps the query vectors $Q$ to outputs using the key-value pairs as follows:

$$\text{Att}(Q, K, V) = \omega(QK^T)V \tag{1}$$

where the dot product $(QK^T) \in \mathbb{R}^{n_q \times n_v}$ and $\omega(\cdot)$ is softmax function. The dot product returns the similarity of each query and key value. The output $\omega(QK^T)V \in \mathbb{R}^{n_q \times d_v}$ is the weighted sum over $V$, where larger weight corresponds to larger similarity between query and key.

A powerful extension to the above (single-) attention mechanism is the multi-head attention introduced in (Vaswani et al., 2017), which allows the model to jointly attend to information from different representation subspaces at different positions. Instead of computing a single attention function, this method first projects $Q, K, V$ onto $h$ different vectors, respectively. An attention

function $Att(\cdot)$ is applied individually to these $h$ projections. The output is a linear transformation of the concatenation of all attention outputs:

$$\text{MultiAtt}(Q, K, V) = \text{concat}(O_1, O_2, \cdots, O_h)W^o,$$
$$O_{i'} = \text{Att}(QW_{i'}^q, KW_{i'}^k, VW_{i'}^v) \text{ for } i' \in 1, \cdots, h, \quad (2)$$

where $W^o, W_{i'}^q, W_{i'}^k, W_{i'}^v$ are learnable parameters of some linear layers. $QW_{i'}^q \in \mathbb{R}^{n_q \times d_q^h}$, $KW_{i'}^k \in \mathbb{R}^{n_v \times d_q^h}$, $VW_{i'}^v \in \mathbb{R}^{n_v \times d_v^h}$ are vectors projected from $Q, K, V$ respectively. $d_q^h = d_q/h$ and $d_v^h = d_v/h$.

Following the architecture of the transformer (Vaswani et al., 2017; Lee et al., 2019), we define the following multi-head attention block:

$$\text{MultiAttBlock}(Q, K, V) = LayerNorm(Q' + Q'W^{q'}), \quad (3)$$
$$Q' = LayerNorm(\text{concat}(QW_1^q, \cdots, QW_h^q) + \text{MultiAtt}(Q, K, V))$$

where $W^{q'} \in \mathbb{R}^{d_q \times d_q}$ is a learnable linear layer. Base on the multi-head attention block, we further define a self-attention block as follows:

$$\text{SA}(X) = \text{MultiAttBlock}(X, X, X) \quad (4)$$

**Contrastive Learning**  Contrastive learning (CL) has been an increasingly popular and effective representation learning approach (Chen et al., 2020; He et al., 2020; Khosla et al., 2020). As the first proposed CL approach, self-supervised CL (Chen et al., 2020) is proposed to learn presentations in an unsupervised manner. Specifically, a mini-batch is constructed from the original input images and their augmented versions. Given a minibatch of $2N$ instances $I = \{1...2N\}$ and an anchor instance $i \in I$, the augmented version of $i$, denoted as $i_a \in I$ is considered the positive sample, and the other $2(N-1)$ instances within the mini-batch are considered negative examples. The loss function of self-supervised CL is defined as follows:

$$\mathcal{L}_{self} = -\sum_{i \in I} \log \frac{\exp(z_i \cdot z_{i_a}/\tau)}{\sum_{z_a \in A(i)} \exp(z_i \cdot z_a/\tau)} \quad (5)$$

where $z_i$ is the image embedding, $z_i \cdot z_{i_a}$ denotes the inner dot product between two embeddings, $\tau \in \mathbb{R}^+$ is a scalar temperature parameter, and $A(i) = I \backslash z_i$. It can be seen that self-supervised CL pushes the embeddings of the samples augmented from the same image close to each other. More recently, the supervised CL (Khosla et al., 2020) adapts CL into the supervised settings, which utilizes the label information to select positive and negative samples. For supervised CL, the embeddings of the samples with the same labels are pushed close to each other, which achieves better performance in classification tasks. For more details of CL, we refer to Liu et al. (2021b). To our knowledge, CL has not been applied in solving multi-label classification yet.

## 2.2 PROPOSED METHOD

Unlike previous CL methods for single-label classification that use image-level representations, we propose to learn multiple label-level representations for each image, which facilitates the application of CL in multi-label classification. We introduce MulCon, which consists of three main neural network modules: the label-level embedding network, the contrastive learning projection network, and the classification networks as shown in Figure 1.

**Label-Level Embedding Network**  As the key module of our framework, the label-level embedding network takes an image $x_i$ as input and outputs its label-level representations, denoted by $g_i \in \mathbb{R}^{L \times D}$, where each row of $g_i$ corresponds to the embedding of the image under the context of a specific label. Specifically, the label-level embedding network consists of two components: 1) *The encoder block.* We first adopt an encoder network that learns the image-level embedding as the backbone model as the backbone model: $r_i = \text{Enc}(x_i) \in \mathbb{R}^{C \times H \times W}$, where $C, H, W$ are the number of channels, height, and width of the output. The backbone model can be implemented with an arbitrary model that learns good image features, e.g., a convolutional neural network (CNN) such as ResNet (He et al., 2016) or visual transformer (Dosovitskiy et al., 2020). We then reshape

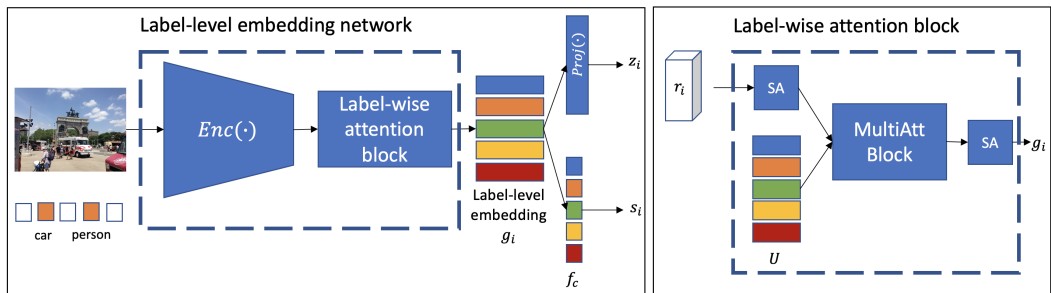

Figure 1: **Left**: Overall structure of our **MulCon** framework. It consists of three main components: label-level embedding network that uses a label-wise attention block with an Encoder (Enc(·)) for extracting label-level embeddings $g_i$ from input images, a set of independent classifiers $f_c$ for multi-label prediction, and a projector (Proj(·)) to map label-level embeddings to a latent space for contrastive learning. **Right**: Detail of the label-wise attention block. It takes image features $r_i$ from (Enc(·)) as input and returns label-level embedding $g_i$ as output. The module contains several self-attention blocks, a multi-headed attention block and a set of learnable label embedding.

$r_i \in \mathbb{R}^{WH \times C}$ for the next step's processing. 2) *The label-wise attention block.* In this block, we introduce a set of vectors $U \in \mathbb{R}^{L \times C}$, each row of which is to be learned as the global embedding for a specific label. To generate an image's label-level embeddings $g_i$, we propose a cascade of several self-attention blocks and a multi-head attention block to capture the interactions between $r_i$ and $U$ as follows:

$$r_i = \text{SA}(r_i); \quad g_i = \text{MultiAttBlock}(U, r_i, r_i); \quad g_i = \text{SA}(g_i). \tag{6}$$

To summarise, given an image $x_i$, we first learn its representation $r_i \in \mathbb{R}^{WH \times C}$ by the encoder, each row of which captures the feature of a location in the image. We then pass $r_i$ to a self-attention block that captures the interactions between the features of the image. Next, we feed $r_i$ to the multi-head attention block, where the global label embeddings $U$ is the query and $r_i$ is the key and value. As the sizes of the input and output channels of the multi-head attention are $C$ and $D$, respectively, we have $n_q = n_v = C$ and $d_q = n_q = D$ for Eq. 2. With the multi-head attention block, our model can learn the "importance" (attention weight) of the image feature to a specific label. For example, if an image contains an apple, the corresponding embedding is expected to be associated with a large attention weight from the apple label. The multi-headed attention can also help a label pay its attention to multiple objects in an image, i.e., each of the attention heads can generate attention scores for a class-specific embedding over all the image-level embeddings. In this case, if an image consists of multiple apples, each of the apples receives a specific attention score from the apple label. Following up the multi-head attention is a self-attention block that implicitly helps improve label correlation. With the attention from all the labels, we can derive the label-level embeddings $g_i$ of the image from its image-label embedding $r_i$.

**Contrastive Learning Projection Network** After obtaining $g_i \in \mathbb{R}^{L \times D}$, we use $g_{ij} \in \mathbb{R}^D$ to denote the label-level embedding of the input image $i$ under the context of a specific label $j$ ($j \in \{1, \cdots, L\}$). Following Chen et al. (2020); Khosla et al. (2020), our framework includes a projection network Proj(·) that maps $g_{ij}$ to a vector in another embedding space: $z_{ij} = \text{Proj}(g_{ij}) \in \mathbb{R}^{d_z}$, where the contrastive learning is performed.

**Classification Network** Recall that the label-level embedding $g_{ij}$ captures the input image $i$'s feature under the context of the label $j$. Thus, it can be used to predict whether $j$ is active in $i$. Accordingly, we introduce a fully connected layer as a classifier $f_c^j$ to predict the probability of the label $j$ being active. Specifically, for each label $j \in L$ the prediction score is $s_{ij} = \sigma(f_c^j(g_{ij})) \in (0, 1)$. We further denote $s_i \in (0, 1)^L$.

### 2.3 LEARNING MULCON WITH CONTRASTIVE LOSS

After introducing the framework, we describe the learning process of MulCon by showing the loss function first. Based on the label-level embeddings of the image $i$, i.e., $g_i$, we introduce a loss

function with two terms: the classification loss and contrastive loss. The overall training process is illustrated in Figure 2.

**Classification Loss** For the classification loss, given the predictive probabilities output from the classification network $s_i$ and the ground-truth multi-hot label vector $y_i$, we apply the binary cross-entropy (BCE) loss, which has been widely used for multi-label classification:

$$\mathcal{L}_{BCE} = \sum_{j=1}^{L} y_{ij} \log s_{ij} + (1 - y_{ij})\log(1 - s_{ij}) \tag{7}$$

It is noteworthy that other multi-label classification losses than BCE can also be used in our framework, e.g., in Ben-Baruch et al. (2020); Lin et al. (2017).

**Label-level Contrastive Loss** In addition to the classification loss, we introduce the label-level contrastive loss (LLCL) for multi-label classification, which is one of the key contributions of this paper. Because an image is associated with multiple labels, it is hard to directly apply CL as we analyzed before. However, in MulCon, after learning the label-level embeddings for an image, we show that the multi-label problem can be transformed into a single-label one, where CL can be adapted in a straightforward way.

As LLCL works in the projected space, hereafter, we also call $z_i$ the label-level embeddings for image $i$, which is projected from $g_i$. Given a minibatch of $N$ images, we first forward-pass them through the label-level embedding network and contrastive learning projection network and then aggregate the label-level embeddings of all the images into set $Z = \{z_{ij} \in \mathbb{R}^{d_z} | i \in \{1, \cdots, N\}; j \in \{1, \cdots, L\}\}$. Similarly, we define the set of the ground-truth labels of the minibatch: $Y = \{y_{ij} \in \{0,1\} | i \in \{1, \cdots, N\}; j \in \{1, \cdots, L\}\}$. If we view an image's label-level embedding $z_{ij}$ as an instance instead of the image itself, $z_{ij}$ is associated with a single ground-truth label $y_{ij}$. We further define $I = \{z_{ij} \in Z | y_{ij} = 1\}$ as the set that contains the label-level embeddings with active ground-truth labels and $A(i,j) = I \setminus z_{ij}$ as the set contains the embeddings in $I$ with $z_{ij}$ excluded.

In the minibatch, we now consider $z_{ij} \in I$ as the anchor, which presents the feature of image $i$ under active label $j$. With LLCL, we aim to push $z_{ij}$ closer to the embeddings under the same active label $j$ of other images in the minibatch, i.e., the positive set, defined as $P(i,j) = \{z_{kj} \in A(i,j) | y_{kj} = y_{ij} = 1\}$. With above notations and inspired by the supervised CL, we define the contrastive loss for the anchor $z_{ij}$ as:

$$\mathcal{L}_{LLCL}^{ij} = \frac{-1}{|P(i,j)|} \sum_{z_p \in P(i,j)} \log \frac{\exp(z_{ij} \cdot z_p / \tau)}{\sum_{z_a \in A(i,j)} \exp(z_{ij} \cdot z_a / \tau))}, \tag{8}$$

The loss for the whole minibatch is: $\mathcal{L}_{LLCL} = \sum_{z_{ij} \in I} \mathcal{L}_{LLCL}^{ij}$. Together with the classification loss, we show the overall training loss of MulCon:

$$\mathcal{L} = \mathcal{L}_{BCE} + \gamma \mathcal{L}_{LLCL}. \tag{9}$$

where the parameter $\gamma$ controls the trade-off between the two losses.

**Why and How does Contrastive Loss Help?** Now we would like to answer why contrastive loss helps in multi-label classification, by showing that it serves as an important complementary to the classification loss. Specifically, with the BCE classification loss applied in many multi-label classification problems, each label can be viewed to be classified independently with a specific classifier as discussed in Section 2.2. That is to say, each classifier focuses on the classification of a specific label and cares less about the *distinctiveness* of the features of different labels. Distinctiveness in classification means that we expect features of the instances with the same label to be close to each other, which has been known as an important factor for achieving good classification accuracy. Similar to single-label problems, the label-level contrastive loss in MulCon is designed to enforce distinctiveness of the label-level embeddings. To demonstrate this, we show the t-SNE (Van der Maaten & Hinton, 2008) visualization of the (active) label-level embeddings of 1,000 randomly sampled images of the COCO dataset (Lin et al., 2014) in Figure 3. These embeddings are from MulCon trained by the BCE loss (i.e., Eq. 7) and the combined loss of BCE and LLCL (i.e., Eq. 9), respectively. Each dot represents one label-level embedding under the context of a specific label and each color represents one class. It can be seen that compared with the loss with BCE only, the additional contrastive loss makes the label-level embeddings of the same label fall into more compact

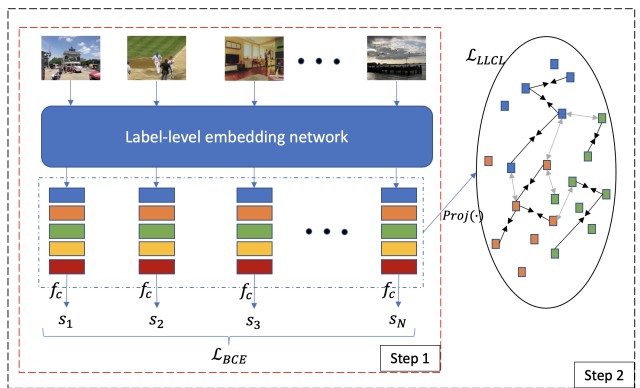

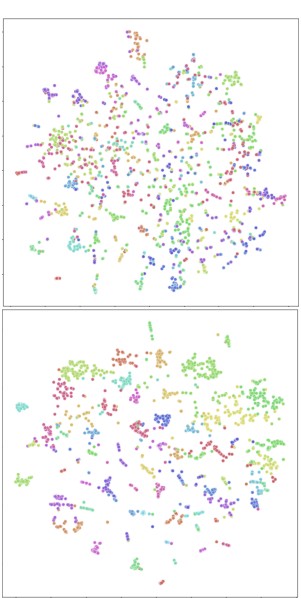

Figure 2: **MulCon** has two steps during training: pretraining and contrastive finetuning. The first step is to train the label-level embedding network with binary cross-entropy loss ($\mathcal{L}_{BCE}$) to effectively decompose an input image into several semantic components so that the first component corresponds to the first label, etc. The second step is to finetune the previously trained network with contrastive loss ($\mathcal{L}_{LLCL}$) and $\mathcal{L}_{BCE}$ to improve the quality of label-level embedding.

Figure 3: t-SNE visualization for image components trained with only $\mathcal{L}_{BCE}$ (top) and with the combination of $\mathcal{L}_{BCE}$ and $\mathcal{L}_{LLCL}$ (down).

clusters, which are better separated from the clusters of other labels. This clearly shows the enforced distinctiveness of the embeddings with the contrastive loss.

However, unlike in single-label classification, being over distinct in the embedding space is not always a good thing in multi-label classification. Specifically, label correlations are important in multi-label classification (Chen et al., 2019c; You et al., 2020). For example, when we see a cup in an image, the probability of a table existing below the cup in the image can be assumed to high. In this case, we actually do not want the embeddings of cups and tables are too far or too separated from each other. Therefore, the enforced distinctiveness with the contrastive loss needs to be judiciously controlled. We propose a simple training strategy to tackle this task, which consists of two steps: *the pre-training step and the contrastive learning step*. In the pre-training step (Step 1), we pretrain the backbone and label-level embedding network and the classification network of MulCon with the BCE loss only. In the contrastive learning step (Step 2), we then plug in the contrastive projection network with LLCL but also keep the BCE loss. In the first step, the BCE loss learns the label-level embeddings freely and implicitly obtains semantic structure by the effect of several attention blocks. After the embeddings are learned, we finetune them with LLCL to enforce distinctiveness of the embeddings. The network is fintuned with small learning rate which helps improve feature distinctiveness without breaking the label semantic structure. We empirically find that the propose training policy works well in practice.

## 3 RELATED WORK ON MULTI-LABEL IMAGE CLASSIFICATION

Our paper mainly focuses on two things: contrastive learning and multi-label classification. To our knowledge, most of the works in contrastive learning are for single-label classification. Thus, although there is an emerging literature of contrastive learning, the details are omitted as our paper solves a different problem. Multi-label classification has been a challenging and important problem in computer vision, where many methods have been proposed. We mainly consider two lines of multi-label methods as our close works.

**Methods capturing label/class correlations** For example, (Wang et al., 2016; Yang et al., 2016; Liu et al., 2017; Chen et al., 2018a) explicitly capture the class correlation by a CNN-based model

followed by a Recurrent Neural Network (RNN) (Medsker & Jain, 2001). Moreover, there are also approaches based on on probabilistic graphical model (Li et al., 2016a; 2014), which model label dependencies in the covariance of probabilistic distributions. Finally, as label correlations can be formulated into graphs, some approaches apply Graph Convolutional Network (GCN) (Zhou et al., 2020) or leverage graph structures to learn label presentations such as in Yang et al. (2016); Chen et al. (2019c); You et al. (2020).(Li et al., 2019) proposes to align image feature with label feature. Compared with these methods, ours is a different approach that controls label distinctiveness and correlation via contrastive loss.

**Methods with attentions** Another research line close to ours is using visual attention for multi-label classification, such as in Guo et al. (2019); Chen et al. (2018b;a); Wang et al. (2017); Ba et al. (2015); Huynh & Elhamifar (2020); Yazici et al. (2020). In addition, attention mechanisms are also introduced in Chen et al. (2019a); Ye et al. (2020) to capture label correlation. Recently, vision transformers (Dosovitskiy et al., 2020) has been a new paradigm of computer vision tasks and there emerges new works that use vision transformers for multi-label classification (Liu et al., 2021a; Lanchantin et al., 2021; Cheng et al., 2021), where attention mechanisms are naturally used. In our method, we use the attention mechanism for a different purpose, i.e., to assist contrastive learning. Specifically, the attention is used to derive an image's label-level embeddings from its image-level embeddings and the global-label embeddings. Moreover, our framework is flexible to also use vision transformers as the backbone model.

# 4 EXPERIMENTS

In this section, we compare our proposed model with the state-of-the-art multi-label classification methods. More results are shown in the appendix. we use the two most popular benchmark datasets: MS-COCO (Lin et al., 2014) and NUS-WIDE (Chua et al., 2009). To evaluate the classification performance, we use the standard metrics for multi-label classification including the mean average precision (mAP) mean average precision (mAP), overall precision (OP), recall (OR), F1-measure (OF1) and per-category precision (CP), recall (CR), and F1-measure (CF1), computed over all prediction scores and top-3 highest prediction scores. As F1 is a comprehensive metric computed by precision and recall, we consider mAP, CF1, and OF1 as more important metrics. The formulas of the metrics are shown in the appendix. For a fair comparison, we adopt ResNet-101 (He et al., 2016) as the backbone for our method to extract image-level features, which is the same for the other compared methods, unless otherwise specified. The implementation and training details of our methods are shown in the appendix.

## 4.1 EXPERIMENTS ON MS-COCO

In the standard setting for multi-label classification, MS-COCO contains 122,218 images with 80 different categories, where every image is associated with 2.9 labels on the average. The dataset is divided into 82,081 images for training and 40,137 images for validation. For a fair comparison, we choose state-of-the-art methods that use Resnet101 as backbone network, and further divide them into two groups. Group 1 uses input image size $448 \times 448$ such as Multi Evidence (Ge et al., 2018a), CADM (Chen et al., 2019b), ML-GCN (Chen et al., 2019c), KSSNet (Liu et al., 2018), MS-CMA (You et al., 2020) and MCAR (Gao & Zhou, 2021). Group 2 uses larger input image size such as SSGRL (Chen et al., 2019a), C-Trans (Lanchantin et al., 2021) and ADD-GCN (Ye et al., 2020).

The results on MS-COCO are reported in Table 1. The numbers of the compared methods are taken from the best reported results in their papers. It can be seen that our approach achieves the state-of-the-art results especially on mAP, CF1 and OF1. For example, in the $448 \times 448$ input resolution with all prediction scores, ours outperforms the second best (MS-CMA and MCAR) by 1.1% on mAP and 0.8% on CF1. With the resolution increased to $576 \times 576$, ours surpasses the second best (ADD-GCN) by 1.1% on mAP and 0.7% on CF1. Although our method may not always achieve better results on precision (CR and OR), it consistently obtains better performance on F1, which is a more comprehensive metric covering both precision and recall.

| Method | Resolution | All | | | | | | | Top-3 | | | | | |
|---|---|---|---|---|---|---|---|---|---|---|---|---|---|---|
| | | mAP | CP | CR | CF1 | OP | OR | OF1 | CP | CR | CF1 | OP | OR | OF1 |
| Multi Evidence | $448 \times 448$ | - | 80.4 | 70.2 | 74.9 | 85.2 | 72.5 | 78.4 | 84.5 | 62.2 | 70.6 | 89.1 | 64.3 | 74.7 |
| CADM | $448 \times 448$ | 82.3 | 82.5 | 72.2 | 77.0 | 84.0 | 75.6 | 79.6 | 87.1 | 63.6 | 73.5 | 89.4 | 66.0 | 76.0 |
| ML-GCN | $448 \times 448$ | 83.0 | **85.1** | 72.0 | 78.0 | 85.8 | 75.4 | 80.3 | **89.2** | 64.1 | 74.6 | 90.5 | 66.5 | 76.7 |
| KSSNet | $448 \times 448$ | 83.7 | 84.6 | 73.2 | 77.2 | 87.8 | 76.2 | 81.5 | - | - | - | - | - | - |
| MS-CMA | $448 \times 448$ | 83.8 | 82.9 | 74.4 | 78.4 | 84.4 | 77.9 | 81.0 | 86.7 | 64.9 | 74.3 | 90.9 | 67.2 | 77.2 |
| MCAR | $448 \times 448$ | 83.8 | 85.0 | 72.1 | 78.0 | **88.0** | 73.9 | 80.3 | 88.1 | 65.5 | 75.1 | **91.0** | 66.3 | 76.7 |
| MulCon (Ours) | $448 \times 448$ | **84.9** | 84.0 | **74.8** | **79.2** | 85.6 | 78.0 | 81.6 | 87.8 | 65.9 | 75.3 | 90.5 | **67.9** | 77.6 |
| SSGRL | $576 \times 576$ | 83.8 | **89.9** | 68.5 | 76.8 | **91.3** | 70.8 | 79.7 | **91.9** | 62.5 | 72.7 | **93.8** | 64.1 | 76.2 |
| C-Trans | $576 \times 576$ | 85.1 | 86.3 | 74.3 | 79.9 | 87.7 | 76.5 | 81.7 | 90.1 | 65.7 | 76.0 | 92.1 | **71.4** | 77.6 |
| ADD-GCN | $576 \times 576$ | 85.2 | 84.7 | 75.9 | 80.1 | 84.9 | 79.4 | 82.0 | 88.8 | 66.2 | 75.8 | 90.3 | 68.5 | 77.9 |
| MulCon (Ours) | $576 \times 576$ | **86.3** | 84.7 | **77.3** | **80.8** | 85.9 | 79.9 | **82.8** | 88.6 | 67.2 | **76.5** | 91.0 | 68.8 | **78.4** |

Table 1: Results on the COCO dataset. The best scores are highlighted in boldface. More important metrics including mAP, CF1, and OF1 are highlighted in grey.

| Method | mAP | CF1 | OF1 |
|---|---|---|---|
| FitsNet | 57.4 | 54.9 | 70.4 |
| attention-transfer | 57.6 | 55.2 | 70.3 |
| s-CLs | 60.1 | 58.7 | 73.3 |
| MS-CMA | 61.4 | 60.5 | 73.8 |
| SRN | 62.0 | 58.5 | 73.4 |
| MulCon (Ours) | **63.9** | **61.8** | **74.8** |

Table 2: Results on NUS-WIDE dataset. The best scores are highlighted in boldface.

| Method | mAP | CF1 | OF1 |
|---|---|---|---|
| R101 + BCE | 80.8 | 76.2 | 79.2 |
| R101 + BCE + SCL | 80.8 | 76.0 | 79.1 |
| LLEN + BCE | 83.8 | 78.8 | 81.1 |
| LLEN + BCE + LLCL | 83.7 | 78.8 | 81.1 |
| MulCon | 84.9 | 79.2 | 81.6 |

Table 3: Ablation study of different variants and training policies of MulCon.

## 4.2 Results on NUS-WIDE

The NUS-WIDE dataset originally contained 269,648 images from Flicker and has been manually annotated with 81 visual concepts. Since some URLs have been deleted, we follow (Ben-Baruch et al., 2020) to obtain 200,000 images with 2.4 labels per image on average. We use the same hyperparameter setting as MS-COCO for training and testing on this dataset. We also select the state-of-the-art methods that report the results on NUS-WIDE, including FitsNet (Romero et al., 2014), attention-transfer (Zagoruyko & Komodakis, 2016), s-CLs (Liu et al., 2018), CMA (You et al., 2020) and SRN (Zhu et al., 2017). Following the standard and convention of many other works, we report mAP, CF1, OF1 computed with all prediction scores. The results on NUS-WIDE are shown in Table 2. It can be seen that MulCon achieves the best results in all the three metrics. For example, it our performs SRN (the second best on mAP) by a significant margin, i.e., nearly 2%.

## 4.3 Ablation Study

To fully understand the modules and training policies of MulCon, we provide a comprehensive ablation study in this section. Specifically, we are interested in the following variants of our method.
**1) R101 + BCE:** The backbone model, Resnet101, trained with the BCE loss (i.e., Eq. 7). This is a standard multi-label classification baseline. **2) R101 + BCE + SCL:** Resnet101 trained with the BCE and supervised-CL (Khosla et al., 2020) (SCL) losses (i.e., Eq. 9). This is a variant where the supervised CL loss is directly applied on the image-level features. Note that for this variant, each image is an anchor, and thus we need to define the positive/negative sets for each anchor. Following the spirit of (Khosla et al., 2020), we consider other images in the minibatch that have at least one common active label with the anchor as its positive samples and the others as the negative ones.
**3) LLEN + BCE:** The label-level embedding network (LLEN) with R101 as the backbone trained with BCE, which corresponds the MulCon framework trained in Step 1 described in Section 2.3. **4) LLEN + BCE + LLCL:** The variant where we train MulCon with the BCE and LLCL losses from the beginning, i.e., the second contrastive learning step described in Section 2.3. **5) MulCon:** The complete model of MulCon trained with the two-step policy described in Section 2.3.

Table 3 shows the results for the ablation study for our method in MS-COCO. We have the following remarks: **1)** By comparing between R101+BCE and R101+BCE+SCL, we can see that unlike in single-label classification, directly applying SCL on the image-level features cannot improve the performance in multi-label cases (80.8 vs 80.8 mAP). **2)** It can be observed that by using the proposed label-level embedding network instead of R101 (R101+BCE V.S. LLEN+BCE), we can can significantly improve the mAP score (80.8 vs 83.8 mAP). **3)** Note that the difference between LLEN + BCE + LLCL and MulCon is on the training policies: The former is trained with BCE and LLCL from the beginning while the later is first pretrained with BCE (Step 1) and then futhur trained by BCE + LLCL (Step 2). We observe that if the LLCL loss is applied from the beginning of the training, it can barely improve the performance. This is because LLCL may excessively enforce distinctiveness which makes the model care less about label correlations, as analyzed in Section 2.3. It can be seen that our proposed two-step training policy improves significantly.

## 4.4 QUALITATIVE ANALYSIS

To qualitatively study our method, we first exam the semantics captured by the label-level embeddings learned with MulCon by conducting an image retrieval experiment. Specifically, given an input image $x_i$, we do a forward-pass to get its label-level embeddings $g_i$. With the ground-truth labels of $x_i$, we can specify a specific active label $j$, pick its embedding $g_{ij}$ from $g_i$ and use it as the query for image retrieval. By comparing the Euclidean distance between the query embedding with the label-level embeddings of other images, we can retrieve the closest images to our query image and label. Figure 4 shows the image retrieval results of MulCon and its variant with BCE only. More qualitative analysis is provided in the appendix.

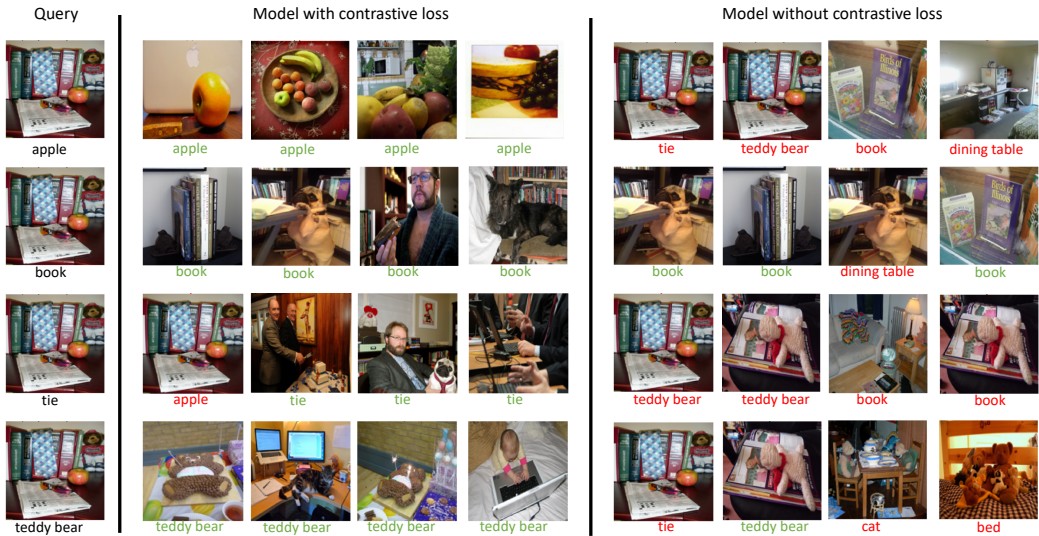

Figure 4: Top-4 related images retrieved given an query image and label on COCO dataset. The results for our full model (MulCon) are on the left, and the results for our model without contrastive loss (MulCon with BCE only) are on the right. The label under each retrieved image is the one corresponding to the embedding closest to the picked query embedding.

## 5 CONCLUSION

In this paper, we have introduced an end-to-end multi-label image classification framework, MulCon, which leverages contrastive learning in multi-label classification. It has been shown that CL is not directly applicable in this domain, due to the fact that it is hard to define the positive/negative samples for an anchor with multiple labels. To tackle this issue, we have introduced to learn label-level embeddings for an image with the multi-head attention mechanism. With label-level embeddings, we transform the multi-label classification into a single case for each label-level embedding,

which facilities a straightforward adaption of supervised CL. We have provided analytical and empirical study of why CL helps in multi-label learning and proposed a simple training policy to control the distinctiveness enforced by CL. Extensive experimental results and visualization show the effectiveness of our approach and its ability to achieve the state-of-the-art performance in multi-label image classification.

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

## A  APPENDIX

In this section, we provide more details on the following subjects:

- Implementation Details of MulCon
- Evaluation Metrics
- More Qualitative Analysis

### A.1  IMPLEMENTATION DETAILS OF MULCON

**Projector**  Following other contrastive learning methods (Chen et al., 2020; Khosla et al., 2020), the projector Proj($\cdot$) has two linear layers with ReLU($\cdot$) activation.

**Step 1: Pretraining with BCE**  We use Adam optimizer (Kingma & Ba, 2015) and 1-cycle policy with the maximum learning rate is 2e-4. The training batch size is set to 64, and the dimension of the label-level feature, i.e., $D$, is 1024.

**Step 2: Finetuning with LLCL**  We use Stochastic Gradient Descent (SGD) optimizer with momentum 0.9 and weight decay is 1e-4. The learning rate is initialized as 0.01 and then reduced by a factor of 10 for every 20 epochs. The batch size for this step is 32. The temperature $\tau$ is 0.2, and $\gamma$ is 0.1. The number of head in multi-head attention is empirically set to 4, which gives us the best results. Following Ben-Baruch et al. (2020), we also employ Cutout (DeVries & Taylor, 2017) with factor 0.5 as regularization. During training, exponential moving average (EMA) is applied to model's parameters with decay factor 0.9997.

**Augmentations in Step 2**  It has been known that augmentations are important for contrastive learning. Following previous methods (Chen et al., 2020; Khosla et al., 2020), we have an augmentation module to transform each training image before feeding them to the network. Specifically, for the augmentation, we first resize the input image to $448 \times 448$, and then apply random horizontal flip, and a Random Augmentation module (Cubuk et al., 2020). Note that for the second step training with batch size 32, the actual batch size becomes 64 after data augmentation.

Note that we apply contrastive loss on label-level embeddings, not on images. For a batch size of 32 images, on average we have 94 label-level embeddings (with the active labels for each image). After augmentation,the batch size of the label-level embeddings is 188 already, which is the maximum batch size that we can run with our computational resource. We will consider larger batch size in future.

### A.2  EVALUATION METRICS

Beyond mean average precision (mAP), the standard metrics reported in the experimental section are: overall precision (OP), recall (OR), F1-measure (OF1) and per-category precision (CP), recall (CR), F1-measure (CF1). These metrics are computed as follows:

$$OP = \frac{\sum_i TP_i}{\sum_i TP_i + FP_i} \qquad OR = \frac{\sum_i TP_i}{\sum_i TP_i + FN_i}$$

$$CP = \frac{1}{C} \sum_i \frac{TP_i}{TP_i + FP_i} \qquad CR = \frac{1}{C} \sum_i \frac{TP_i}{TP_i + FN_i}$$

$$OF1 = \frac{2 \times OP \times OR}{OP + OR} \qquad CF1 = \frac{2 \times CP \times CR}{CP + CR}$$

where $TP_i$ is true positive of class $i$, $FP_i$ is false positive of class $i$, $FN_i$ is false negative of class $i$.

### A.3  MORE RESULTS ON VOC AND VG500

**Experiments with VOC 2007**.

We compare our method with RDAL(Wang et al., 2017), RARL(Chen et al., 2018b), MCAR(You et al., 2020), SSGRL(Chen et al., 2019a) and ASL (Ben-Baruch et al., 2020) on

VOC2007(Everingham et al., 2015) dataset. Following other works, we also use pretrained backbone on COCO and fine-tune on VOC 2007. Overall, our method has a comparable results with other methods. We also try to use TResnet and ASL loss combine with our Contrastive framework, but we do not observe any improvement. Figure 4

| Method | aero | bike | bird | boat | bottle | bus | car | cat | chair | cow | table | dog | horse | mbike | person | plant | sheep | sofa | train | tv | mAP |
|---|---|---|---|---|---|---|---|---|---|---|---|---|---|---|---|---|---|---|---|---|---|
| RDAL | 98.6 | 97.4 | 96.3 | 96.2 | 75.2 | 92.4 | 96.5 | 97.1 | 76.5 | 92.0 | 87.7 | 96.8 | 97.5 | 93.8 | 98.5 | 81.6 | 93.7 | 82.8 | 98.6 | 89.3 | 91.9 |
| RARL | 98.6 | 97.1 | 97.1 | 95.5 | 75.6 | 92.8 | 96.8 | 97.3 | 78.3 | 92.2 | 87.6 | 96.9 | 96.5 | 93.6 | 98.5 | 81.6 | 93.1 | 83.2 | 98.5 | 89.3 | 92.0 |
| MCAR | 99.7 | **99.0** | 98.5 | 98.2 | 85.4 | 96.9 | 97.4 | **98.9** | 83.7 | 95.5 | 88.8 | 99.1 | 98.2 | 95.1 | 99.1 | 84.8 | 97.1 | **87.8** | 98.3 | 94.8 | 94.8 |
| SSGRL | 99.7 | 98.4 | 98.0 | 97.6 | 85.7 | 96.2 | 98.2 | 98.8 | 82.0 | 98.1 | 89.7 | 98.8 | 98.7 | 97.0 | 99.0 | 86.9 | 98.1 | 85.8 | 99.0 | 93.7 | 95.0 |
| ASL | **99.9** | 98.4 | 98.9 | **98.7** | **86.8** | 98.2 | **98.7** | 98.5 | 83.1 | 98.3 | 89.5 | 98.8 | **99.2** | **98.6** | **99.3** | **89.5** | 99.4 | 86.8 | 99.6 | 95.2 | **95.8** |
| MulCon | 99.8 | 98.3 | **99.3** | 98.6 | 83.3 | **98.4** | 98.0 | 98.3 | **85.8** | **98.3** | **90.5** | **99.3** | 98.9 | 96.6 | 98.8 | 86.3 | **99.8** | 87.3 | **99.8** | **96.1** | 95.6 |

Table 4: Results on VOC07. Best results are highlighted in boldface.

**Experiments with VG 500**.

We compare the performance on VG 500 (Krishna et al., 2017) with other methods such as R101(He et al., 2016), R101-SRN(Zhu et al., 2017), SSGRL(Chen et al., 2019a) and C-Trans(Lanchantin et al., 2021) in Table 5. Our method also achieve a better results compare to other methods.

| Method | mAP |
|---|---|
| R101 | 30.9 |
| R101-SRN | 33.5 |
| SSGRL | 36.6 |
| C-Trans | 38.4 |
| MulCon | 38.5 |

Table 5: Results on VG 500.

### A.4 CLASS EMBEDDING RETRIEVAL

We compute the precision@k (P@k) and recall@k (R@k) of class embedding retrieval between our method and Q2L (Liu et al., 2021a) in Table 6. We use the first 1000 images in MS-COCO for this experiment. We extract all the class embedding to create the set $X \in R^{N \times d}$, N is the total number of embeddings and d is the embedding's dimension. Then for an embedding $x_i$, we construct a groundtruth label vector $y_i$ such that $y_{ij} = 1$ if $x_i$ and $x_j$ has the same label, $y_{ij} = 0$ otherwise. Then we have the ground truth set $Y \in R^{N \times N}$. Given $X$, we can compute the pair-wise cosine similarity $C \in R^{N \times N}, c_{ij} = cossim(x_i, x_j)$. Given $C$ and $Y$, the P@k and R@k is computed as follows:

$$P@k = \frac{TP@k}{TP@k + FP@k}$$
$$R@k = \frac{TP@k}{min(k, TP@k + FN@k)}$$

where $TP@K, FP@k, FN@k$ are true positive, false positive and false negative at top k predictions.

| Method | P@3 | P@5 | P@10 | P@15 | P@20 | R@3 | R@5 | R@10 | R@15 | R@20 |
|---|---|---|---|---|---|---|---|---|---|---|
| MulCon | 87.6 | 84.7 | 80.6 | 76.9 | 73.1 | 87.7 | 84.7 | 80.7 | 77.4 | 74.8 |
| Q2L | 77.7 | 73.7 | 69.5 | 66.3 | 62.6 | 77.8 | 73.8 | 69.6 | 66.7 | 64.1 |

Table 6: Quanntitative results of class embedding retrieval on a subset of MS-COCO.

| Query | Closest labels | | |
|-------|--------|---------|-------|
| person | handbag | cellphone | chair |
| fork | oven | knife | sink |
| donut | hotdog | pizza | sandwich |

Table 7: Label correlations on COCO.

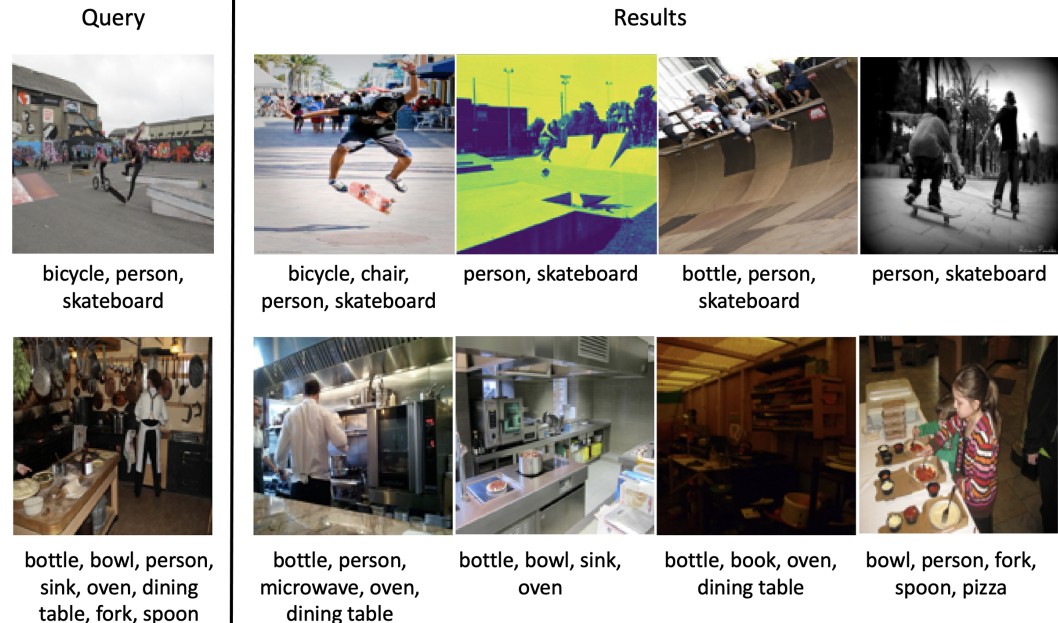

Figure 5: Top-4 related images retrieved given an query image and multiple labels on COCO dataset.

### A.5 MORE QUALITATIVE ANALYSIS

**Multi-label Image Retrieval** To further demonstrate the intuitive meanings of the learned label-level embeddings in the multi-label setting, we provide an additional retrieval experiment where instead of using a single query label, we select multiple ones and concatenate their label-level embeddings as the query vector. That is to say, given an image, we can retrieve the images with multi-label labels of interest. Figure 5 shows the multi-label retrieval results.

**Visualisation of Attention Maps** We now show a visualization for the attention maps in terms of the active labels of an image produced by the multi-head attention used in the label-level embedding network. As illustrated in the top row of Figure 6, the attention maps can precisely highlight the regions of the image in terms of each of its ground-truth labels. In the bottom row of Figure 6, we show the attention maps of the heads of one label "person". It can be observed that the heads tend to capture the multiple instances of "person" in the image. Figure 7 presents more attention maps visualization. Here we visualize the mean attention map computed from 4 heads.

**Analysis on the Global Label Embeddings** A row of $U$ is the embedding vector of a specific label, encoding the semantic information of the label. Although randomly initialised, the embedding vectors are learned by minimising the loss function via gradient backpropogation. In Table 7, we have computed the pairwise Euclidean distances between label embeddings and the closest 3 labels retrieved with minimum distances. It can be seen that the label correlations are quite meaningful.

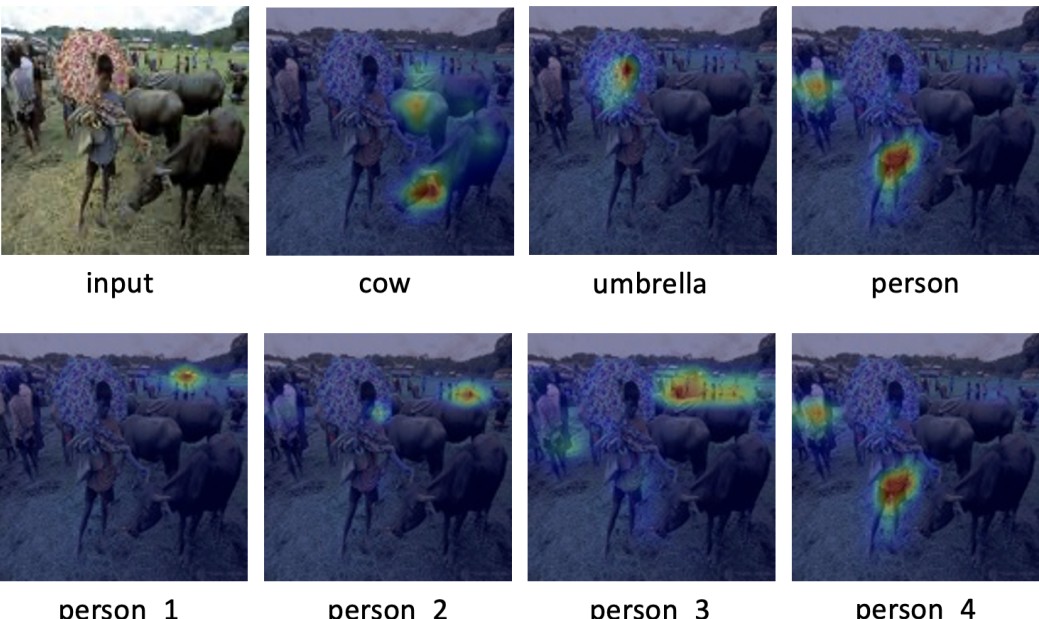

Figure 6: Visualization of attention maps. The top row includes the input image and the selected attention map for the ground-truth labels. The bottom row includes the multi-headed attention maps for class "person".

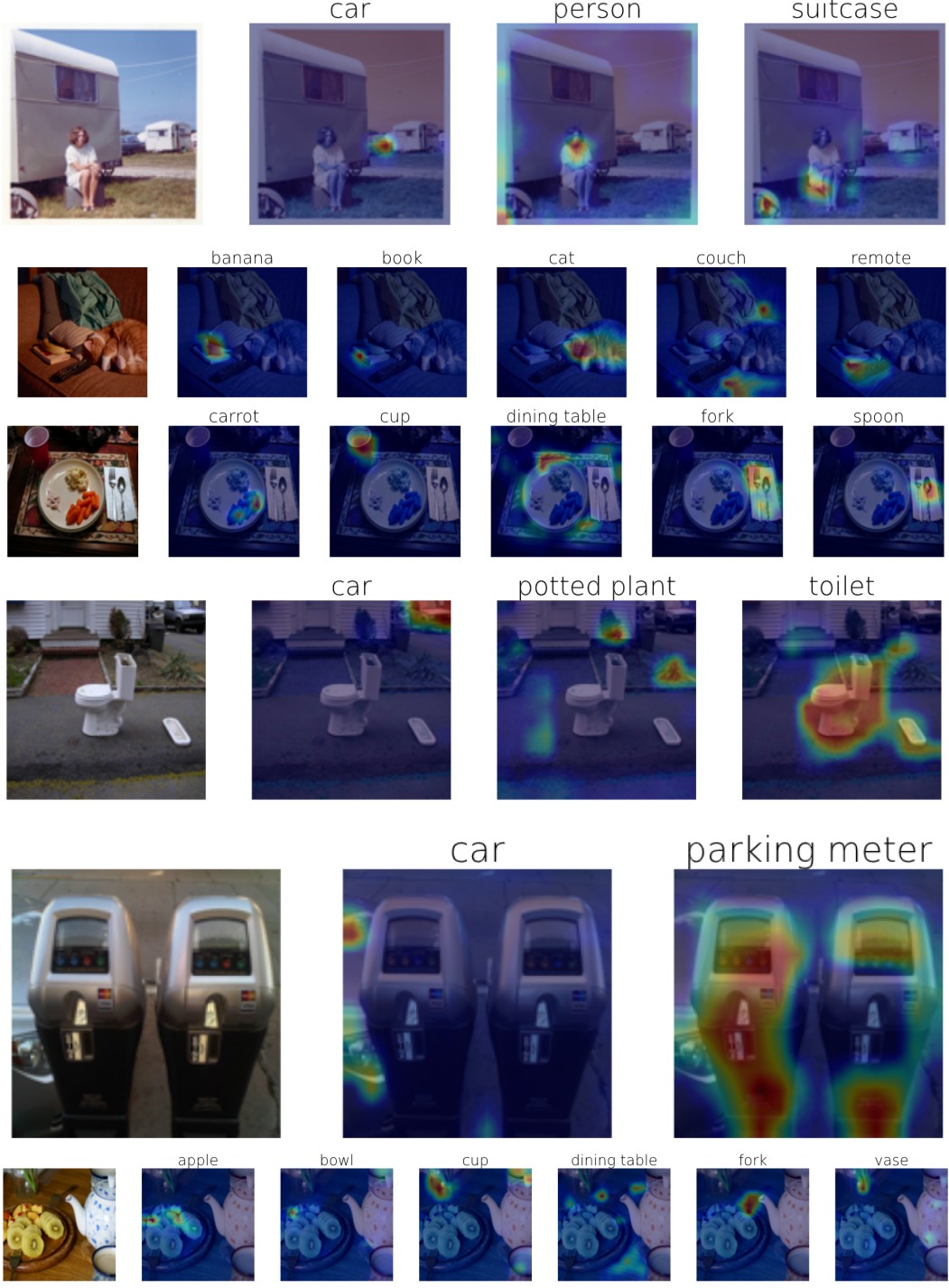

Figure 7: Visualization of attention maps. In each row, the left most image is input image, and the remains are the mean attention maps (from 4 head) for its ground truth classes.

