# OpenReview forum: "Contrastively Enforcing Distinctiveness  for Multi-Label Classification"
_ICLR.cc/2022/Conference — ICLR 2022 Submitted_

### Official Review · Reviewer_rsBB · 2021-10-29

**Correctness:** 3
**Technical Novelty And Significance:** 2
**Empirical Novelty And Significance:** 2
**Recommendation:** 5
**Confidence:** 4

**Main Review:**

Strengths:
1. The paper is written very well and easy to follow.
2. The proposed approach is technically sound and achieves the new state-of-the-art.
3. Interesting qualitative analysis for the properties of learned representations.

Weaknesses:
1. It is an incremental work and the main focus, i.e. the adaptation of contrastive learning for multi-label classification, is a trivial generalization of existing contrastive learning method.
2. There are some exist works which introduce the idea of contrastive learning into multi-label classification. Please talk about the relationships between these existing works and the approach in this paper. Experimental comparisons with some of these works would be desirable.
3. According to the results reported in the ablation study (LLEN+BEN+LLCL vs. LLEN+BEN), it seems that the contrastive learning term may not be a good regularization for multi-label learning. While the proposed MulCon trained with a two-step policy has a much better performance, I am really confused about whether the performance gain comes from the contrastive learning term or the improved training procedure, since many additional training tricks are utilized. More analyses are suggested to uncover the true sources of the performance gain.


There are some related works which introduce the idea of contrastive learning into multi-label classification:

[1] Li, C., Liu, C., Duan, L., Gao, P., & Zheng, K. Reconstruction regularized deep metric learning for multi-label image classification. TNNLS, 2019.

[2] Chen, C., Wang, H., Liu, W., Zhao, X., Hu, T., & Chen, G. Two-stage label embedding via neural factorization machine for multi-label classification. AAAI, 2019.

[3] Liu, W., & Tsang, I. W. Large margin metric learning for multi-label prediction. AAAI, 2015.


**Summary Of The Paper:**

In this paper, the authors introduce the contrastive learning into multi-label classification. Specifically, the multi-label classification problem is first decomposed into a series of binary classification problems with label-level features extracted by the attention mechanism. Then, label-wise contrastive learning is performed on these binary classification problems respectively. Comparative experiment shows the proposed approach achieves the new state-of-the-art performance in multi-label image classification.

However, the proposed adaption of contrastive learning for multi-label classification is a trivial generalization of existing contrastive learning for single-label classification, since contrastive learning is simply performed on the binary classification problem of each class label. And it is not a new idea to introduce the contrastive learning into multi-label classification. Besides, extracting label-level features via attention mechanism is a well-established technique in many existing works. Thus, if I had not missed something, the contribution of this paper is very limited.


**Summary Of The Review:**

Considering the limited contributions and unclear sources of the performance gain, I think this paper is marginally below the acceptance threshold.

---

> ### Author Response · Authors · 2021-11-21
> **Response to reviewer 3**
>
> **1. Novelty of the method.**
>
> Please refer to our response to all the reviewers.
>
> **2. Compare with related works.**
> Thanks for pointing out several related papers.
> We have put them in the references and updated the paper (in blue). We would like to provide the following discussions on these papers.
>
> a) Discussions.
>
> In high-level ideas, there are connections between contrastive learning and metric learning, which are briefly discussed in the paper "A simple framework for contrastive learning of visual representations", the most popular paper in contrastive learning from Hinton's group. In our opinion, contrastive learning can be viewed as a specific way of metric learning, which is done by using positive and negative samples.
>
> Despite such connections, our method is fundamentally different from the three papers.
>
> - For these methods, they use metric learning approaches to enforce the embeddings of labels to be close to the embeddings of features. While ours is to push the visual embeddings of a category (label) across different images to be close to each other. That is to say, they are pushing images to labels in the latent space while ours is pushing images themselves together.
>
> - Ours specifically works for multi-label image classification. [2,3] are for general multi-label classification problems. It requires more study on how effective these methods are when adapted to image classification.
>
> - The architecture and implementation of ours are also different from these methods including [1].
>
> b) Empirical comparisons.
>
> We believe that we have compared most of the state-of-the-art published methods in the area of multi-label image classification. But we agree that more comparisons can make our paper more comprehensive.
>
> Please note that [2,3] are not specifically for image classification. Therefore, it would be hard for us to directly compare the two methods.
> [1] is for multi-label image classification and also reports on the CoCo dataset that was used in our experiments. We can compare the proposed method, named RETDM, with ours on CoCo. The results are in the following table. We use the same setting for the input image (224 * 224 resolution) as in RETDM.
>
> | Method     | CP | CR  | CF1 | OP | OR | OF1|
> | :---        |    :----:   |   :----: |   :----: |   :----: |   :----: |   :----: |
> | RETDM[1]      | 80.0       | 55.5   |65.5 | 81.9 | 61.1 | 70.0 |
> | MulCon   |  82.1     | 66.3   |73.4 | 82.3 | 70.6 | 76.0 |
>
> **3. Ablation study.**
>
> Please allow us to provide further explanations which are expected to address the confusion here.
>
> a) To examine whether the proposed contrastive loss helps, in Table 3 of the main paper, we shall focus on LLEN + BCE, LLEN + BCE + LLCL, MulCon (LLEN + BCE + LLCL + two-stage training policy).
>
> b) In our study, we found that directly applying LLCL (contrastive loss) to the whole training stage can make the features of different labels over distinct (That is what contrastive loss is designed for). However, this is not always a good thing in multi-label classification. Please refer to our detailed analysis at the end of Section 2.3 (Why and How does Contrastive Loss Help?).
>
> c) Therefore, we propose a simple two-stage training policy, which is: We first train by LLEN + BCE and then we use LLEN + BCE + LLCL. There might be a misunderstanding on this point. Please note that the only thing changed in the second stage is adding the contrastive loss. That is to say, no additional training tricks are used except the proposed LLCL. We believe that the contrastive loss is the true source of the performance gain.

---

> > ### Comment · Reviewer_rsBB · 2021-11-24
> > **Response to rebuttal**
> >
> > Thanks for your responses.
> >
> > The added discussions on related works are pertinent and further experiments are convincing.
> >
> > Nevertheless, I still think contributions of this paper is limited. And my concerns on the ablation studies still exit, since additional training tricks (Cutout, EMA, etc.) are used in the finetune stage as reported in the appendix A.1. Thus, I would like to keep my score.

---

### Official Review · Reviewer_ToSu · 2021-11-01

**Correctness:** 3
**Technical Novelty And Significance:** 3
**Empirical Novelty And Significance:** Not applicable
**Recommendation:** 6
**Confidence:** 4

**Main Review:**

Multi-label classification is one of the fundamental problems in computer vision and has a large amount of studies in the literature. This paper  addressed the problem by contrastive learning, which has not been explored for multi-label classification. The proposed solution that learns label-level embeddings is intuitive and reasonable, and the results are promising.

The paper is well organized and written. It also provides sufficient amount of background to understand the proposed method.

However, I have two questions for further clarification:

1. The proposed label-level embedding network is considered one of the main contributions in this work. This network receives an image and returns a set of label-level embeddings. The architecture of this network involves attention blocks, similar to visual transformer and its variants. In fact, Query2Label [1] adopts transformer (taking advantage of cross-attention modules as well) and achieves a similar performance level. What is exactly the difference between the proposed design and those works?

[1] Liu et al., Query2Label: A Simple Transformer Way to Multi-Label Classification, arXiv:2107.10834.

2. The proposed method achieved state-of-the-art performance on both MS COCO and NUS-WIDE datasets; however, these two datasets have different characteristics. For example, COCO contains some "small objects" (e.g., spoon, cell phone, etc.) and NUS-WIDE has a few "concepts" (e.g., map). Image embeddings learned by contrastive embedding and attention mechanism may help recognition of small objects, but why is it helpful for recognizing concepts that are global, and perhaps somewhat vague? Does the performance gain against existing methods come mainly from the attention module?


**Summary Of The Paper:**

This paper presents a contrastive learning based method for multi-label classification. In particular, authors propose to learn "label-level embeddings" for an image, thereby the multi-label classification problem can be transformed into a single-label one where contrastive learning can be naturally adapted. The proposed label-level embedding network involves self-attention and multi-head attention blocks that learn label-specific embeddings. Results were demonstrated by the experiments on two benchmark datasets (MS COCO and NUS-WIDE).



**Summary Of The Review:**

This paper presents new state-of-the-art results on MS COCO and NUS-WIDE for multi-label classification. It leverages contrastive learning to enhance distinctiveness for great performance in multi-label image classification. It is also well organized and written.

The proposed label-level embedding network is conceptually similar to visual transformer or other attention variants. I am not sure about the novelty at current point. The results were demonstrated on MS COCO and NUS-WIDE only. I am not sure if it is scalable to larger datasets (like open images dataset).

---

> ### Author Response · Authors · 2021-11-21
> **Response to reviewer 2**
>
> **1. Comparisons with Query2Label.**
>
> Please refer to our response to all the reviewers.
>
> **2. "Small objects" and "concepts"**
>
> Thanks for pointing out this interesting point. Our thinking is as follows:
>
> a) We would consider that the proposed contrastive loss is general enough to help in both cases. In general, we can assume that even in different images, the objects or concepts of the same kind of category shall share some common characteristics. That is why these objects or concepts are categorized into one category.
> Both small objects like apples share similar visual features and so are bigger concepts like maps. Our contrastive loss is pushing the label-level embeddings of a category across different images to be close to each other. Therefore, we believe that our contrastive loss is beneficial to multi-label image classification in general.
>
> b) In the ablation study in Table 3 of the main paper, we can compare the variant of ours without contrastive loss (LLEN + BCE) and the one with contrastive loss (MulCon). Both methods use the same attention architectures, however, the one with contrastive loss outperforms the other significantly. Therefore, we would consider the performance gain is from the use of contrastive loss.
>
> c) In addition to CoCo and NUS-WIDE, we have also conducted new experiments on another two datasets VOC07 and VG500 shown in Table 4 and Table 5 in the appendix, where the labels of VOC07 are objects and these in VG500 consist of both objects and attributes.
> It can be seen that our method also achieves the state-of-the-art performance in the two datasets.

---

> > ### Comment · Reviewer_ToSu · 2021-11-30
> > **Response to authors' comments**
> >
> > Thank you for your feedback. My previous questions are answered. I will keep my score.

---

### Official Review · Reviewer_2oF2 · 2021-11-02

**Correctness:** 3
**Technical Novelty And Significance:** 2
**Empirical Novelty And Significance:** 3
**Recommendation:** 5
**Confidence:** 4

**Main Review:**

Strengths:
- Built upon the transformer encoder-decoder structure, authors manage to obtain the label-level embedding for further contrastive learning, of which the label-level embedding among one mini-batch construct a natural training set for contrastive constrains;
- The in-depth investigation have been made to analyze why and how dose contrastive loss help, including a vivid t-sne illustration;

Weaknesses:
The idea of conducting the contrastive learning within the mini-batch among the label-level embedding is interesting. However, the major concern is the overall novelty of proposed method.
- The attention-based label-level embedding is a straightforward application of transformer-like structure, which have been shown in the recent multi-label classification models;
- Apart from the scenario of training with network, the bag-of-words kind training policy is also quite common in multi-label classification; holding the contrastive learning as the main contribution of the paper is quite limited;
- The experimental results are still less advanced compared to the recent sota models, such as query2label, TResNet, MlTr etc., which can be referred at paper with codes (https://paperswithcode.com/sota/multi-label-classification-on-ms-coco). It is known that the multi-label performances are sensitive to input size, backbone, and training strategies etc., authors still need to expand their experiments to fully validate the effectiveness of the contrastive learning;
- Also only two datasets are investigated, what about VOC07/12, Visual Genome? As the proposed framework is highly related to the label-level embedding, it should be verified on different label-set size to investigate its limitations.

**Summary Of The Paper:**

In this paper, authors propose a framework for utilizing the contrastive learning to improve the feature distinctiveness for multi-label classification. The main purpose is to address that the direct application of contrastive learning (similar to the single-label classification case) is unhelpful for improving the multi-label performance.

**Summary Of The Review:**

Overall, the idea of utilizing the contrastive loss during the training of multi-label image classification network is appealing, however the limited novelty and experiments are the main concerns. Authors should carefully address above concerns (weaknesses) during their feedback, the final recommendation will be made upon the feedback.

---

> ### Author Response · Authors · 2021-11-21
> **Response to reviewer 1**
>
> Thank you very much for the valuable feedback. Our response to the comments are as follows:
>
> **1,2. Novelty and significance of the proposed method**
> Please refer to our response to all the reviewers.
>
> **3. Settings of input sizes, backbones, training strategies, etc.**
> We completely agree that these settings affect multi-label classification results. Therefore, we tried our best to control these settings to get fair comparisons with other methods. Specifically, we used two settings of the input sizes (Table 1 in the main paper) and for all the compared methods, we applied the same backbone model (i.e., ResNet101). Moreover,
> except the proposed contrastive loss and the two-step training strategy, we kept other settings the same for all the methods in comparison. Therefore, we believe that the performance gain is from the proposed contrastive loss.
>
> Thanks for pointing out methods like query2label, TResNet, MlTr. We are confident that with proper adaptions, our method can also be beneficial for these bigger models. We are currently running the experiments. However, given the model sizes, and time and computational resource limits, we have not got the results yet. We will try our best to report in the rebuttal phase.
>
> **4. Results on more datasets**
> We agree that it is better to report on more datasets. Originally, we reported on COCO and NUS-WIDE, which we believe are two of the most widely-used datasets in multi-label image classifications. Following your suggestion, we have tested our method on VOC07 and Visual Genome (VG500) and report the comparison with others in Table 4 and Table 5 in the appendix of the updated paper.
> It can be observed that in these two datasets, our proposed method outperforms or is comparable to the state-of-the-art methods.

---

> > ### Comment · Reviewer_2oF2 · 2021-11-28
> > **Response to author comments**
> >
> > Thanks for addressing my concerns regarding the experimental settings and additional results, it is very much appreciated that authors conduct the extended experiments to further verify the contributions.
> > However, my main concerns are still the limited novelty and the generalization ability of the given method. Apart from the contrastive learning, the model shares the similar structure with the sota frameworks, and the proposed learning strategies performs marginally above the compared models especially when it is investigated on more datasets, which shows the limited generalization ability. Therefore, I would like to keep my score.

---

### Official Review · Reviewer_3XCq · 2021-12-06

**Correctness:** 3
**Technical Novelty And Significance:** 3
**Empirical Novelty And Significance:** 3
**Recommendation:** 6
**Confidence:** 3

**Main Review:**

Strengths:
+ The organization of this paper is good, and it's easy to read.
+ The proposed method seems to be an intuitive adaptation of the contrastive learning formula to multi-label classification.
+ Experimental evaluation proves the attractiveness of the proposed method. ResNet-101-based MulCon beats other baselines on most of the presented measures. I believe that using only ResNet-101 models is a fair way of conducting such a comparison and four datasets are enough.
+ The appendix seems to consist of enough details to reproduce these results.
+ Ablation study (including results from discussion), qualitative analysis and pretty t-SNE visualization.

Weaknesses and doubts:
- The scope of the contribution is a bit limited, it seems to me that the main novelty in the area of multi-label classification is proposed Label-level Contrastive Loss. The rest is pretty much a simple arrangement of known blocks and techniques.
- The motivation for the usage of contrastive learning is shallow.
The major performance boost comes from the attention block, which, as I understand, has almost the same number of parameters as ResNet-101 used as base-encoder. I lack a comment on a number of parameters of the proposed method and baselines in the paper.
- There is no information about the maximum numbers of epochs for each MulCon training step. I suspect that two-step training of MulCon may take more time than other benchmarks since finetuning with LLCL that use SGD may require a large number of epochs + MulCon has a lot of additional parameters.
- Is augmentation described in the appendix is also used with R101 + BCE + SCL, and LLEN + BCE + LLCL variants in the ablation study presented in Table 3? It is not clear from the paper. One of the authors' responses suggests that it was not. It is also not clear which other techniques from the finetune phase are necessary for this framework to work.

Nits:
- It's not difficult to guess (based on results), but for clarity, it would be nice to add resolution used for ablation study.
- Different ways of writing ResNet-101/Resnet101, it would be nice to unify the form.

**Summary Of The Paper:**

The authors introduce a new framework for multi-label classification that leverages supervise contrastive learning. The framework adds an attention mechanism on top of the image encoder. Thanks to that, per-label features can be obtained to perform label-wise contrastive learning. The proposed method (MulCon) is trained in two steps. The first step performs pre-training by optimizing only binary-cross entropy (BCE) loss. In the second step, the model is finetuned using a combination of BCE and proposed by the authors Label-Lebel Contrastive Loss. The experimental study on four datasets shows that the proposed approach achieves the new state-of-the-art results among other methods based on ResNet-101, on most from many performance measures.

**Summary Of The Review:**

The review was written after the rebuttal phase, and authors' responses to other reviewers' comments were taken into account. I find this paper to be a sold work, a simple idea with good empirical evaluation. In my opinion, the authors also did a good job answering the reviewers' comments. However, I still have some other doubts, so I believe that right now, this paper is marginally above the acceptance threshold - I recommend it for acceptance, but I will not be upset if the paper is rejected.

---

> ### Author Response · Authors · 2021-12-06
> **Thanks for the review**
>
> We thank the reviewer for the valuable and insightful comments, and we understand that additional reviews are usually required to be submitted in a short period of time. Therefore, the reviewer's effort is more appreciated. Our response is as follows:
>
> **1. Novelty**
>
> Contrastive learning is a general idea, concept, and framework for learning data features in unsupervised or supervised problems. It has the potential to be applied in various areas. To our knowledge, it has not been carefully studied in multi-label image classification. In this paper, we aim to leverage the success of contrastive loss (especially the supervised contrastive loss) in single-label classifications into multi-label classifications. This is a straightforward and intuitive idea, but it is underdone in the literature to our knowledge. To accommodate contrastive loss in multi-label image classification, we developed a conceptually simple model with the right building blocks and proposed effective training policies accordingly. We believe that these efforts are non-trivial. We would also consider simplicity as an advantage of our proposed method.
>
> **2. Motivation of contrastive loss**
>
> The introduced contrastive loss is well-motivated both conceptually and empirically. Conceptually, we provided the detailed discussion in  "Why and How does Contrastive Loss Help?"  of Section 2.2 in the main paper. Empirically, we agree that compared with the vanilla R101, the attention block does provide some performance gain. But the contrastive loss can further improve the performance significantly.
>
> To further show how the contrastive loss helps, we have provided a better-organised table for ablation study with additional results, shown in the following table.
>
> | Method  |   mAP | CF1 | OF1|
> | :---        |    :----:   |   :----: |   :----: |
> | R101 + BCE  | 80.8  | 76.2 | 79.2  |
> | LLEN + BCE   | 83.8 | 78.8 | 81.1|
> | LLEN + BCE + Cutout + EMA + RandAug  |  84.0 | 79.1 | 81.3 |
> |LLEN + BCE + LLCL + Cutout + EMA + RandAug (MulCon) | 84.9 | 79.2 | 81.6 |
>
> Please note that the first, second, and fourth rows are copied from Table 3 in the main paper and the third row is from the new experiments.
>
> We have the following remarks:
> - In contrastive learning, the training techniques including Cutout, EMA, and RandAug are important and widely used, because contrastive learning is about pushing an image and its augmentations close in the feature space.
> - Although these techniques can also be used to improve non-contrastive learning methods, the improvement is reported much less significant. This is can be seen by comparing LLEN + BCE V.S.  LLEN + BCE + Cutout + EMA + RandAug in the above table, noting that both variants are without the contrastive loss. That means that using these training techniques only does not contribute to the performance gain significantly.
> - If we compare LLEN + BCE + Cutout + EMA + RandAug with the proposed MulCon, the only difference is the proposed contrastive loss. It confirms that the contrastive loss is the main source of the performance gain, i.e., nearly 1%.
>
> About the number of parameters, with $576\times 576$ resolution in Table 1 of the main paper, we have R101 (44.5 million) and ours (86.8 million). However, C-Trans (Lanchantin et al., 2021), a recent strong baseline with attentions as well, has 120.4 million parameters. That is to say, ours achieves better performance than C-Trans with a smaller architecture.
>
> **3. More information for training**
>
> We set the maximum number of epochs for the second step of MulCon to 100. However, to get the best performance on the validation set, we have 70 epochs for COCO, about 20-30 epochs for NUS-WIDE, and 10 epochs for VG 500 and VOC 2007, respectively. We believe these are not large numbers for training with SGD.
>
> **4. Augmentation**
>
> Reviewer rsBB also commented on this point. Yes, it was our bad that we did not show the ablation study clearly before. We have provided a better-organised table for ablation study with additional results, shown in the table in the response to the second comment "Motivation of contrastive loss".
>
> With the results in the table, we can confirm that the performance gain of our method comes from the contrastive loss, not from the augmentation or other training tricks.
>
> We also responded similarly to Reviewer rsBB. Although we have not got Reviewer rsBB's reply yet, we believe that the new table is convincing and hope it to address your and Reviewer rsBB's concerns.
>
> **5. Nits**
>
> Thanks a lot for these. We will definitely address them.

---

### Comment · Area_Chair_kUgH · 2021-11-10
**Discussion phase**

Dear Authors and Reviewers,

Let me first thank you for supporting ICLR 2022: Authors for submitting their contributions, and Reviewers for going through them and sending their comments and remarks!

As the discussion phase has just begun, let me ask Authors to answer all questions appearing in reviews and to defend your paper, and reviewers to check all other reviews to see whether you coincide and to be ready to respond to authors rebuttals.

We are also looking forward for public comments. I hope for a vivid discussion for this paper.

Best regards, AC for Paper 1395

---

### Author Response · Authors · 2021-11-21
**Common concerns from reviewers**

We thank all the reviewers for the valuable feedback.
Besides the responses to the specific comments from individual reviewers, we would also like to highlight the following key points in our rebuttal.

**1. Novelty and significance of the proposed method**

Contrastive learning is a general idea, concept, and framework for learning data features in unsupervised or supervised problems. It has the potential to be applied in various areas. To our knowledge, it has not been carefully studied in multi-label image classification.

In this paper, we aim to leverage the success of contrastive loss (especially the supervised contrastive loss) in single-label classifications into multi-label classifications. This is a straightforward and intuitive idea, but it is underdone in the literature to our knowledge. Moreover, we believe that it is non-trivial to implement and make it work effectively in practice, which requires a series of significant adaptations and explorations in terms of model architectures and training policies.

Finally, please kindly note that we are not developing a new network architecture for multi-label classification but adapting/proposing an appropriate application of contrastive loss. We claim this is our main focus, novelty and significance.

**2. Comparison with Query2Label [1]**

Thanks to the reviewers who pointed out the Query2Label (Q2L) work. We have the following remarks in the comparison with Q2L.

a. Please note that to our knowledge, Q2L was introduced in an Arxiv paper at the end of July 2021 and has not been published in formal venues yet. At the time we conducted our work, the paper was not released. Therefore, we would consider Q2L as a concurrent paper with ours. However, we are happy to discuss our thoughts on comparing Q2L with ours.

b. In this paper, we aim to leverage the success of contrastive loss (especially the supervised contrastive loss) in single-label classifications into multi-label classifications. Please kindly note that we are not developing a new network architecture for multi-label classification but adapting/proposing an appropriate application of contrastive loss. Therefore, the novelty of model architectures is not the focus. In terms of architecture, both Q2L and ours use multi-head attentions and backbone image encoders to learn label level features. However, Q2L uses several transformer layers which have much more parameters than the multi-head attentions used in our method. For example, on COCO with image resolution $448 \times 448$ and ResNet101 as the backbone, our model has a comparable performance with Q2L. However, our model has 86.8 million parameters, while Q2L is with 193.9 million parameters, which are more than double ours. We believe that the use of contrastive learning in our model helps it get similar performance with Q2L with a much smaller architecture.

c. One of the main advantages of the proposed contrastive loss is to push the label level embeddings of the same category of different images to be close to each other. As shown qualitatively in Figure 4 in the main paper, the benefit of contrastive loss can be demonstrated by the image retrieval task.
To quantitatively compare in this task, we have conducted a new experiment on image retrieval shown in subsection Class Embedding Retrieval of the appendix of the main paper.
It can be observed that our model significantly outperforms Q2L thanks to the use of contrastive loss.

[1] Liu et al., Query2Label: A Simple Transformer Way to Multi-Label Classification, arXiv:2107.10834.

---

### Decision · Program_Chairs · 2022-01-20

**Decision:**

Reject

**Comment:**

This is an interesting paper trying to answer how contrastive learning can be used in multi-label classification. The reviewers however had raised several doubts about motivations, novelty, or the impact of contrastive module on final results. For many of them the authors had delivered satisfying responses, but after a long discussion, we decided that the paper needs revision to improve in these aspects. For example, the authors should make it clear whether the image retrieval application from Section 4.4 is the main motivation of the method. If so, what are the competitive approaches to solve such problem? How to measure the performance of such methods? Answers for the above questions are crucial to find the right motivations for the contrastive module used by the authors.

We hope that the authors will follow the recommendations and resubmit the paper to another top conference.